# A Plant-Derived Antigen–Antibody Complex Induces Anti-Cancer Immune Responses by Forming a Large Quaternary Structure

**DOI:** 10.3390/ijms21165603

**Published:** 2020-08-05

**Authors:** Deuk-Su Kim, Yang Joo Kang, Kyung Jin Lee, Lu Qiao, Kinarm Ko, Dae Heon Kim, Soon Chul Myeung, Kisung Ko

**Affiliations:** 1Department of Medicine, Therapeutic Protein Engineering Lab, College of Medicine, Chung-Ang University, Seoul 06974, Korea; emrtn86@gmail.com (D.-S.K.); yangjoo33@naver.com (Y.J.K.); Friend-kj@hanmail.net (K.J.L.); qiaolu@naver.com (L.Q.); 2Department of Stem Cell Biology, School of Medicine, Konkuk University, Seoul 06974, Korea; knko@kku.ac.kr; 3Department of Biology, Sunchon National University, Sunchon 57922, Korea; dheonkim@scnu.ac.kr; 4Department of Urology, Chung-Ang University, College of Medicine, Seoul 06974, Korea; uromyung@cau.ac.kr

**Keywords:** antigen-antibody complex, colorectal cancer, large quaternary structure, plant crossing, vaccine

## Abstract

The antigen–antibody complex (AAC) has novel functions for immunomodulation, encouraging the application of diverse quaternary protein structures for vaccination. In this study, GA733 antigen and anti-GA733 antibody proteins were both co-expressed to obtain the AAC protein structures in a F1 plant obtained by crossing the plants expressing each protein. In F1 plant, the antigen and antibody assembled to form a large quaternary circular ACC structure (~30 nm). The large quaternary protein structures induced immune response to produce anticancer immunoglobulins G (IgGs) that are specific to the corresponding antigens in mouse. The serum containing the anticancer IgGs inhibited the human colorectal cancer cell growth in the xenograft nude mouse. Taken together, antigens and antibodies can be assembled to form AAC protein structures in plants. Plant crossing represents an alternative strategy for the formation of AAC vaccines that efficiently increases anticancer antibody production.

## 1. Introduction

Antigen–antibody complexes (AACs) have been shown to regulate immune responses [1,2]. Immunization using AACs has been used to enhance immunogenicity through the production of specific antibodies against immunogenic epitopes [3]. Although factors that affect the efficacy of AACs that influence immune responses are not fully understood, numerous potential mechanisms have been proposed [4]. The Fc region of AAC can increase antigen uptake via the Fc receptor (FcR) that is present on antigen-presenting cells (APCs) and deliver the antigen for cross-presentation [5,6]. For this process, the roles of the Fc region of antibody have been reported, and they include interactions with both inhibitory FcγRIIB and activating FcγR found on the dendritic cells (DCs) and macrophages [7]. Fc, derived from the immune complex, can recognize various effector cells of the innate immune system [8]. Targeting a vaccine to both DCs and antigen-specific B cells for both MHC class I and MHC class II-restricted processing might generate a more diverse and robust immune response against pathogens and cancer cells [9]. Antibody-derived immunomodulation induced by the formation of large protein complexes results in the induction of beneficial or protective immune responses [10].

Research regarding the immune complex-mediated enhancement of immunization has been proactively conducted using animal cell systems [11,12]. However, the use of plants has garnered attention as an alternative expression system for biopharmaceutical recombinant protein products, such as vaccines and antibodies [13,14,15,16]. The plant expression system, which is a eukaryotic system similar to the animal cell system, is expected to express, fold, and assemble therapeutic glycoproteins as a promising alternative with product safety and economic benefit. Furthermore, plant genetic engineering allows for the modification of the protein structure through glycosylation to increase the effectiveness of the vaccine [17]. Thus, plant-based vaccine candidates have been extensively studied for their potential to enhance immune responses [18,19,20]. Recent studies reported that, as compared to the individual antigens, immune complexes via the expression of an antigen fused to the C-terminus of the heavy chain (HC) and light chain (LC) in animal expression systems enhance the activation of DCs and the overall immune response [11,21,22]. However, it remains to be determined whether the co-expression of antigens and antibodies in plants can induce the assembly of both proteins, resulting in large quaternary AACs that enhance immune responses. In plant, immune complexes have been expressed as an antigen fused to the C-terminus of heavy chain (HC) and light chain (LC) forms [23].

In this study, we demonstrated another way to make immune complex form using the transgenic plant crossing process. Thus, in the present study, transgenic plants expressing the antigen GA733 (an epithelial cell adhesion molecule (EpCAM) highly expressed in colorectal cancer cells) fused to Fc (GA^P^) [24,25,26], and the anti-colorectal cancer mAb CO17-1A recognizing the antigen GA733 (CO^P^) [27,28,29,30,31,32] were crossed to express both the antigen and antibody in F1 plants. The GA^P^ and CO^P^ complexes expressed in the plant were assessed for their expression, structure, and in vitro and in vivo immune functions as a vaccine.

## 2. Results

### 2.1. Generation of F1 Plants Carrying both GA^P^ and CO^P^ Genes

Transgenic plants carrying GA^P^ and CO^P^ HCK and LC genes were obtained by Agrobacterium-mediated transformation [25,26,27,28,29,30,31,32]. The GA^P^ encoding gene was under the control of the enhanced cauliflower mosaic virus (CaMV) 35S promoter (Ca2p) and the tobacco etch virus 5′ leader sequence (TEV). CO^P^ HCK and LC were expressed from Ca2p and Pin2p promoters, respectively. F1 seeds were obtained by crossing the plants equally expressing GA^P^ and CO^P^ (Figure 1), and were germinated in the presence of kanamycin to obtain the F1 plant line #4 equally expressing both GA^P^ and CO^P^ for AAC formation (Figure 1b).

### 2.2. Confirmation of GA^P^ and CO^P^ Gene Insertion and Protein Expression in F1 Plants

PCR was conducted to confirm the insertion of GA^P^ and CO^P^ (HCK and LC) genes in the F1 plants (Figure 2a). Analysis of the F1 plants revealed DNA bands of expected sizes (1,483, 1,471, and 764 bp). PCR bands for both GA^P^ and CO^P^ (HCK and LC) genes were detected in the F1 plants #4, 6, and 11. For the F1 plants #1, 2, 3, and 10, only the band corresponding to GA^P^ gene was detected; however, for the F1 plants #9 and 12, only the bands corresponding to CO^P^ (HCK and/or LC) gene were observed. Western blotting was conducted in order to confirm GA^P^ and CO^P^ protein expression in F1 plants (Figure 2b,c). GAP protein (65 kDa) was detected in the F1 plants #4, 6, 11, and 3 using an anti-human Fc fragment IgG–HRP, whereas no protein was detected in the F1 plant #9 (Figure 2b). HCK (50 kDa) and LC (25 kDa) of CO^P^ were detected in the F1 plants #4, 6, 11, and 9 using an anti-mouse HC and LC IgG–HRP, whereas no protein was detected in the F1 plant #3 (Figure 2c). When compared to the other F1 plants, protein band density for F1 plant #4 was relatively stronger. No bands corresponding to HCK or LC were detected in the non-transgenic and parental GA^P^ transgenic plants. The band corresponding to GA^P^ was not observed in the non-transgenic and parental CO^P^ transgenic plants.

### 2.3. Infrared Fluorescent Western Blotting

From the GA^P^ × CO^P^ plants, the proteins that were purified from leaves showed three major bands having the expected sizes for GA^P^ (65 kDa) and CO^P^ HCK and LC (50 and 25 kDa, respectively) (Figure 2d). Infrared fluorescent blotting detected both GA^P^ and CO^P^ (HCK and LC) purified from the F1 plants (Figure 2e). The detection of GA^P^ and CO^P^ (HCK and LC) proteins using IRDye 800 CW (green) and 680 LT (red) was consistent with their detection by western blotting using the anti-human Fc and anti-mouse IgG antibodies, respectively. The Fc fragment protein band (35 kDa) was observed in the proteins that were extracted from the GA^P^ and GA^P^ × CO^P^ plants (Figure 2d,e asterisk).

### 2.4. Interaction of GA^P^ and CO^P^ Proteins with the Anti-GA Antibody and GA

The interactions between GA^P^ and CO^P^ proteins (from the GA^P^ × CO^P^ plants) with the anti-GA antibody and GA, respectively, were analyzed to determine the binding activities and affinity constants while using surface plasmon resonance. It was speculated that, if GA^P^ and CO^P^ assembled to form AACs (Figure 1b), CO^P^ itself in the AAC could interfere with the binding of the anti-GA antibody and GA^P^. On GLC chips immobilized with GA and the mammalian-derived anti-GA mAb (CO^M^) and CO^P^, higher peaks were observed when compared to those that were observed for both GA^P^ + CO^P^ and GA^P^ × CO^P^ when the same concentration was used (Figure 3a,f). CO^M^ and CO^P^ showed similar interactions with GA on the chip. However, binding capacity of the anti-GA antibody for GA at the dissociation section was slightly higher with CO^P^ as compared to that with CO^M^. As the concentration of GA^P^ in the in vitro mixture (GA^P^ + CO^P^) increased (0, 1, 2.5, 5, and 10 ratio), the peaks gradually decreased, and the dissociation speed increased (Figure 3b). To analyze the interactions between the mammalian-derived GA733 antigen fused to Fc (GA^M^), GA^P^, GA^P^ + CO^P^, and GA^P^ × CO^P^ with CO^P^, the GLC chips were immobilized with CO^P^ (Figure 3c,g). Among all four groups, the highest interaction was observed between GA^M^ and GA^P^ (Figure 3c). GA^P^ + CO^P^ and GA^P^ × CO^P^ both showed lower peak signals than GAM and GA^P^. GA^P^ × CO^P^ showed almost no peak signal with CO^P^ coated on the chip. On the chip that was coated with CO^P^, the peak value decreased with increasing GA^P^ concentration (2:0 and 1:0). When GA^P^ was mixed with CO^P^ at GA^P^: CO^P^ ratios of 2:1 or 1:1, the peaks were lower as compared to those that were observed with GA^P^: CO^P^ ratios of 2:0 or 1:0 (Figure 3d).

### 2.5. Binding Activity of GA^P^ × CO^P^ to the Anti-GA Antibody

CO^M^ and CO^P^ were used as the anti-GA capture antibodies during sandwich ELISA in order to confirm the binding activity of GA^P^ in GA^P^ × CO^P^ plants with an anti-GA antibody (Figure 3e). The binding activities of the capture antibody (COM) differed according to the antigen group [GA^P^, GA^P^ + CO^P^ (*in vitro* mixture of GA^P^ and CO^P^), and GA^P^ × CO^P^]. The absorbance value of GA^P^ and GA^P^ × CO^P^ was significantly higher than that of GA^P^ + CO^P^. In addition, as compared to GA^P^, all of the tested GA^P^ × CO^P^ concentrations showed broad and significantly higher peaks (Figure 3e). Particularly, at low concentrations (62.5 and 125 ng), higher absorbance was observed for GA^P^ × CO^P^ than for GA^P^ and GA^P^ + CO^P^. The GA^P^ + CO^P^ group showed a lower signal when compared to the other groups. The trends for binding to CO^P^ and CO^M^ as capture antibodies were similar.

### 2.6. Large Quaternary Protein Structure for GA^P^ × CO^P^

We hypothesized that GA^P^ and CO^P^ assembled to form large quaternary linear or circular structures, based on the ELISA and SPR results which revealed that compared to GA^P^, GA^P^ × CO^P^ showed relatively higher and lower signals, respectively (Figure 3h,i). The structures of GA^P^ and GA^P^ × CO^P^ were visualized using electron microscopy to understand whether both GA^P^ and CO^P^ actually assembled to form large quaternary protein complexes in the F1 plant (Figure 4a,b). GA^P^ showed the predicted Y-shaped structure with dimerization of two GA^P^ proteins. EM indicated that GA^P^ × CO^P^ formed large, circular quaternary structures of ~20–30 nm in size. Structural differences between GA^P^ and GA^P^ × CO^P^ were confirmed using an Atomic Force Microscope (AFM), in which measurements focused on a random section (Figure 4c,f). Sectional profiles showed GA^P^ to be ~10–15 nm in diameter (Figure 4c). The line in profile image indicates that the size of GA^P^ was similar to that of the Y-shaped mAb (Figure 4d). The diameter of GA^P^ × CO^P^ was ~25–35 nm (Figure 4f,g), which was consistent with the electron microscopy results (Figure 4a,b). Furthermore, dynamic light scattering (DLS) was used to estimate the size distribution of GA^P^ and GA^P^ × CO^P^ (Appendix A). The analogous size distribution of GA^P^ was < 30 nm (Appendix A), whereas that of GA^P^ × CO^P^ was >30 nm (Appendix A). The size and distribution of the purified protein sample GA^P^ × CO^P^ were compared with GA^P^ and CO^P^ using size-exclusion chromatography-high performance liquid chromatography (SEC-HPLC). The SEC-HPLC profiles showed that GA^P^ × CO^P^ contained a mixture of molecular species ranging within two main peaks (Figure 4g). In GA^P^ × CO^P^, one peak was detected at 5–6 min. and, which was not observed in both GA^P^ and CO^P^ (Figure 4g). In GA^P^ and CO^P^, only one peak was observed at 7–8 min.

### 2.7. GA^P^ × CO^P^ Glycosylation Profile

Mass analysis was performed in order to compare the *N*-glycan profile of GA^P^, CO^P^, and GA^P^ × CO^P^ (Figure 5a). As expected, all of the samples presented oligomannose glycans (Man 7–9). CO^P^ presented mainly glycan structures with Man 7, whereas GA^P^ presented Man 7–9 oligomannose glycan structures. Like the CO^P^ and GA^P^, GA^P^ × CO^P^ presented oligomannose glycan structures. Moreover, the relative ratio of Man 7 and 8 oligomannose glycan structures in GA^P^ × CO^P^ were similar to that of Man 7 and 8 glycan structures combined from both GA^P^ and CO^P^.

### 2.8. Immunological Response of Mice Administered with GA^P^ × CO^P^

Using SPR, specific responses for the anti-GA antibody were observed and compared for the mice immunized with GA^M^, GA^P^, GA^M^ + CO^M^, GA^P^ + CO^P^, or GA^P^ × CO^P^ (Figure 5b). When compared to all other groups, a significantly higher peak was observed in the serum samples of mice that were immunized with GA^P^ × CO^P^ (Figure 5b). The peaks for serum samples of the GA^M^, GA^P^, and GA^M^ + CO^M^ group mice were similar. When compared to all the other groups, the peak that was observed in serum samples from mice injected with 1 × PBS was the lowest.

### 2.9. Analysis of Interleukin (IL)-4 and IL-10 in CD4^+^ Th2 Cells

Functional consequences of GA^P^ × CO^P^ immunization were determined by analyzing its cumulative effects on cytokine production. IL-4 and IL-10, secreted by a CD4^+^ subset of Th2 cells, were measured to understand T cell activation in each treatment group (GA^M^, GA^P^, GA^M^ + CO^M^, GA^P^ + CO^P^, and GA^P^ × CO^P^) using the spleen supernatant of immunized mice (Figure 5c,d). The levels of IL-4 and IL-10 were significantly higher in the GA^P^ × CO^P^ immunized mice as compared to those in the 1 × PBS, GA^M^, GA^P^, GA^M^ + CO^M^, and GA^P^ + CO^P^ immunized mice (Figure 5c,d). When compared to the other groups, higher levels of IL-4 and IL-10 in the GA^P^ × CO^P^ group indicated an increase in T cell activation.

### 2.10. Antitumor Activity of Serum Samples from the GA^P^ × CO^P^ Immunized Mice against Human Colorectal Tumors in Nude Mice

In the SW620 colorectal cancer xenograft nude mouse model, after injection with serum from the mice immunized with 1× PBS, GA^M^, GA^P^, GA^M^ + CO^M^, or GA^P^ × CO^P^, the first tumors appeared eight days after cancer cell injection. Thereafter, compared to all the other groups, the tumor size increased more rapidly in the mice injected with 1 × PBS (Figure 6). After 10 days, tumors grew significantly faster in the mice that were treated with sera from GA^M^, GA^P^, and GA^M^+CO^M^ treated mice (S (GA^M^), S (GA^P^), and S (GA^M^+CO^M^), respectively) as compared to those treated with serum from GA^P^ × CO^P^ [S (GA^P^ × CO^P^)] and CO^P^ (mAb as a positive control). By day 15, the mean tumor volume in mice treated with S (GA^P^ × CO^P^) was significantly lower than that in the mice treated with S (GA^M^), S (GA^P^), and S (GA^M^ + CO^M^). The effects of S (GA^P^ × CO^P^) appeared to be similar to those of mAb^P^ CO. When compared to the mice injected with 1× PBS, the final mean tumor size in all groups was significantly different (*p* < 0.01).

## 3. Discussion

In this study, we identified the expression and formation of a human colorectal cancer AAC (GA^P^ × CO^P^), a large protein with quaternary structure, in plant tissues. We also demonstrated that this complex was capable of inducing the production of anti-colorectal cancer antibody that inhibited in vivo tumor growth. Plant cross-fertilization was used as the strategy to express recombinant antigens and antibodies, which resulted in the formation of large quaternary structures in the F_1_ plant.

The F_1_ plants expressing both GA^P^ and CO^P^ were successfully generated via cross-fertilization of plants expressing each transgene. The F_1_ plants were screened using PCR, and the expression of both GA^P^ and CO^P^ genes was confirmed in these plants. Protein lysates from the GA^P^ × CO^P^ plants showed proteins with sizes and expression patterns similar to those that were observed in the lysates from GA^P^ and CO^P^ plants, thereby indicating that the GA^P^ and CO^P^ proteins were independently expressed and purified with no signs of interference observed between expressions of the two transgenes.

GA (Figure 3a,b) or the anti-GA mAb (Figure 3c,d) was coated on the SPR chip to confirm the binding activity of anti-GA mAb to GA^P^ in the GA^P^ × CO^P^ lysates. On the SPR chip coated with GA, the kinetic signals of GA^P^ × CO^P^ and GA^P^ + CO^P^ samples were significantly lower when compared to those of the CO^P^ and CO^M^ samples. In addition, on the chip coated with anti-GA mAb, the signal for GA^P^ × CO^P^ samples was lower than that for the GA^P^ + CO^P^ samples. Furthermore, the reduction of signal in the GA^P^ + CO^P^ sample, along with the increase in GA^P^ concentration, indicated the binding of anti-GA mAb to GA^P^ in the GA^P^ × CO^P^ samples. These results suggested that CO^P^ binds GA^P^ in the GA^P^ × CO^P^ tissues, and that GA^P^ and CO^P^ were tightly assembled to form large quaternary structures in the GA^P^ × CO^P^ plants. When compared to the GA^P^ + CO^P^ samples, the GA^P^ × CO^P^ samples showed a lower signal, which suggested that efficient formation of a large quaternary structure containing GA^P^ and CO^P^ occurred in the GA^P^ × CO^P^ plants.

CO^M^ and CO^P^ were coated on an ELISA plate, and GA^P^, GA^P^ × CO^P^, and GA^P^ + CO^P^ lysates were added to the plate, in order to confirm that GA^P^ in the GA^P^ × CO^P^ samples was bound to the anti-GA mAbs. When compared to the GA^P^ and GA^P^ + CO^P^ samples, the absorbance of the GA^P^ × CO^P^ samples did not decrease; in fact, it increased as the concentration decreased (350, 125, and 62.5 ng). It was hypothesized that due to CO^P^ binding, GA^P^ × CO^P^ would not have any sites available for recognition by the anti-GA antibody. The absorbance of the GA^P^ × CO^P^ sample did not decrease when compared to that of GA^P^, suggesting that the formation of a large quaternary protein structure occurred in GA^P^ × CO^P^ (Figure 3h,i). Absorbance of the GA^P^ + CO^P^ sample decreased compared to that of GA^P^ due to CO^P^-mediated blocking of anti-GA mAb binding, which eventually decreased the absorbance. We speculated that GA^P^ and CO^P^ assembled to form AAC, eventually forming dimeric symmetrical structures and large linear or circular quaternary complex structures (Figure 1b). For AACs, anti-GA mAb-coated ELISA plates can retain GA^P^ from the complex, and the human Fc (from GA^P^) can be recognized by secondary antibodies conjugated to HRP, which eventually resulted in enhanced absorbance (Figure 3h,i). Thus, we speculated that GA^P^ × CO^P^ assembled to form large quaternary structures.

However, based on SPR and ELISA results, it was difficult to determine whether the GA^P^ × CO^P^ complex structure was linear or circular. The EM data indicated that the complex formed circular shapes of 25–30 nm (Figure 4a,b). The size of the GA^P^ × CO^P^ complex was confirmed by AFM and SEC-HPLC analyses, which suggested that the GA^P^ × CO^P^ sample proteins formed large circular complex quaternary structures. The SEC-HPLC indicated that the large quaternary and smaller size complex forms existed in GA^P^ × CO^P^, while the individual GA^P^ or CO^P^ had only dimer forms. Circular protein complexes (30 nm in size) can diffuse rapidly to the lymph nodes, allowing the antigens to be efficiently presented to the B and T cells [33,34]. Thus, it is assumed that large quaternary AACs can enhance immunogenicity, which is essential for vaccine production.

Among the five treatment groups (GA^M^, GA^P^, GA^M^+CO^M^, GA^P^ + CO^P^, and GA^P^ × CO^P^), we found significantly higher immune responses in the GA^P^ × CO^P^-treated mice compared to the GA^M^, CO^P^, and GA^M^+CO^M^-treated mice. This enhanced response resulted in elevated serum antibodies that significantly inhibited human colorectal tumor growth in nude mice. IL-4 and IL-10 are involved in inducing humoral immunity and they play a role in inducing IgG_1_ antibodies, which mediate complement-dependent cytotoxicity (CDC) and recruit effector cells for antibody-dependent cellular cytotoxicity (ADCC) as a promising antibody isotype for tumor immunotherapy [35,36].

In this study, no adjuvant was used for vaccination. Thus, AACs act as self-adjuvants in the induction of an immune response [17]. The benefits of increased antigen presentation via antibody immunomodulation include the potential to modify cytokine expression by APCs, enhance germinal center formation, and elicit strong recall responses [37]. We observed that, when compared to the other groups, the IL-10 levels (associated with macrophages) were higher in the serum of mice treated with the quaternary AAC structure (GA^P^ × CO^P^), thereby suggesting the possibility for the future development of an AAC-based vaccine. Although the mechanism of enhancing the immune response by AAC is not fully understood, several explanations have been suggested, including FcR-mediated upregulation via antigen uptake, the exposure of cryptic epitopes due to antibody binding, and enhanced germinal center formation triggering strong recall responses [38]. DCs can preferentially uptake antigens with diameters of 20–300 nm (virus-like size) by phagocytosis or macropinocytosis [39]. Indeed, the GA^P^ × CO^P^ group proteins formed a circular protein complex of approximately 30 nm, similar to the size of virus particles, which could mimic viral pathogenesis in plants. Although the extent to which the structure of GA^P^ × CO^P^ resembles native viral particles is unclear, immunogenicity studies in mice showed that the GA^P^ × CO^P^ protein complex significantly elicited immune responses, which resulted in the generation of anti-GA antibodies. It seems that the uptake of AACs results in multiple epitopes on the cell surface, resulting in increased antigen presentation and processing in APCs. Furthermore, it was revealed that the Fc region and oligomannose glycans on the complex could interact with FcR and oligomannose receptors, respectively, allowing the AAC to be efficiently presented to APCs [40]. Thus, the large quaternary structure of AAC with Fc and oligomannose glycans facilitates targeting towards relevant APCs for the induction of T cell-mediated immune responses [41,42,43]. These results were consistent with those of previous studies demonstrating that immune complex structures with oligomerization of antigenic proteins, elicited considerably higher antibody responses compared to that elicited by simple antigens, by acting as self-adjuvants [44]. Although 30~40% of proteins were AAC form, 30–40% AAC forms induced more immune responses to produce anti-GA antibodies to inhibit tumor growth. Thus, it is believed that the GA^P^ × CO^P^ complex forms obtained from plant is a promising strategy for obtaining the AAC. In our current study, we showed the immune response to generate anticancer IgGs to inhibit colorectal cancer cell growth in animal model. However, we did not show the protective or preventive immune responses in animal model after the vaccination of the GA^P^ × CO^P^. In the future, it is required to determine protective immunity of GA^P^ × CO^P^ against cancer induced in animal model with its vaccination [45]. In addition, in vitro migration or wounding healing assays would be essential to investigate the anti-metastasis activity of the anticancer IgGs induced in the vaccinated mice [29,46].

In conclusion, cross-fertilization using transgenic plants independently expressing antigens and antibodies can generate F_1_ plants, resulting in the formation of quaternary immune AACs. AACs (GA^P^ × CO^P^) appear to be considerably superior immunogens when compared to the antigens (GA^P^) alone. This effective quaternary structure complex vaccine platform could be used for developing other vaccine candidates using plant systems.

## 4. Materials and Methods

### 4.1. Cross-Fertilization

The stamens of transgenic plants (*Nicotiana tabacum* L. cv Xanthi) expressing either GA^P^ or CO^P^ were removed before they blossomed and CO^P^ pollen were collected [25,27]. Pollen was applied to the stigma of a transgenic plant expressing GA^P^, and the seeds were obtained from fertilized plants. The seeds were sterilized with 10% chlorax and 20% ethanol and washed three times with distilled water. The seeds were germinated on Murashige and Skoog (MS) medium [agar (6 g/L; Duchefa, Haarlem, Netherlands), MS (4.8 g/L; Sigma, St.Louis, MO, USA)] containing kanamycin (100 mg/L). The seeds were grown at 22 °C under a growth chamber. F_1_ plants were grown up to the well-developed root stage in a greenhouse with an average 16 h light/8 h dark photoperiod.

### 4.2. PCR Amplification

Genomic DNA was isolated from the leaves of plants expressing GA^P^ and CO^P^, and F_1_ plants (GA^P^ × CO^P^) while using a DNA extraction kit (Qiagen, Valencia, CA, USA). PCR amplification was performed using primer pairs for GA^P^ and CO^P^ (HC and LC) [25,27]. DNA from a non-transgenic plant was used as the negative control, and pGEM T-easy vectors (Promega, Madison, WI, USA) containing the GA^P^ and CO^P^ genes were used as the positive controls.

### 4.3. Western Blotting

Leaves (100 mg) were homogenized in 300 μL PBS (1×). Western blotting was performed, as described previously [26]. The membranes were incubated in the blocking buffer with the mouse anti-GA antibody (1:500; R&D systems) for 1.5 h at room temperature (RT) and incubated for 1.5 h at RT with the goat anti-mouse IgG conjugated with horseradish peroxidase (HRP; Jackson ImmunoResearch, West Grove, PA, USA) diluted (1:5000) in the blocking buffer in order confirm GA^P^ expression. To detect CO^P^ expression, the membranes were incubated with the blocking buffer containing the goat anti-mouse IgG conjugated with HRP. Protein bands were visualized by exposing the membrane to an X-ray film (Fuji) using a chemiluminescent substrate (Pierce Biotechnology, Waltham, MA, USA).

### 4.4. Purification of Recombinant Proteins from Plants

Plant leaves (approximately 300 g) were homogenized in 1 L extraction buffer (37.5 mM Tris–HCl (pH 7.5), 15 mM EDTA (pH 8), 50 mM NaCl, 75 mM sodium citrate monobasic anhydrous, and 0.2% sodium thiosulfate) and centrifuged at 15,000× *g* for 30 min. at 4 °C. Subsequently, purification was performed, as described previously [26,28].

### 4.5. Infrared Fluorescent Western Blot

Infrared fluorescent western blotting was conducted to confirm the expression of both GA^P^ and CO^P^ in plants. Purified recombinant proteins mixed with 4 μL of 5× loading buffer were electrophoresed using 10% SDS-polyacrylamide gels and then transferred onto nitrocellulose membranes (Millipore, Billerica, MA, USA) using the Mini-Protean II^TM^ system (Bio-Rad, Hercules, CA, USA). Membranes were incubated in TBS buffer for 4 h at RT, followed by incubation with the goat anti-human IRDye 800 CW and goat anti-mouse 680 LT (1:15,000; LI-COR, Lincoln, NE, USA) in the blocking buffer at RT for 1.5 h to detect GA^P^ and CO^P^, respectively. After washing four times for 5 min. each in 1 × TBS at RT, the membranes were scanned using the Odyssey^TM^ CLx infrared imaging system (LI-COR).

### 4.6. SPR

SPR was performed using the ProteOn XPR36 surface instrument (Bio-Rad). GA^M^ (R&D systems) or CO^M^ was immobilized on the GLC sensor chip (Bio-Rad) while using the amine coupling chemistry, as described in the manufacturer’s manual. The resonance units (RU) were approximately 1600–1800. Chip stabilization was performed in PBS-T buffer at a flow rate of 100 μL/min. for 60 s. Each sample (15 μg/mL) was applied to immobilized receptors at pH 6, with a flow rate of 50 μL/min. at 25 °C. After each measurement, the surface of the sensor chip was regenerated using phosphoric acid. In all experiments, data were adjusted to zero and the standard channel. The dissociation and rate constants were calculated using the Proteon Manager (Bio-Rad).

### 4.7. Sandwich ELISA

The MaxiSorp 96-well microplates (Nunc) were coated with 50 μL per well of carbonate–bicarbonate buffer containing 5 μg/mL of CO^M^ and CO^P^ at 4 °C overnight. The plate was washed with 1× PBS and blocked with 1× PBS containing 3% BSA overnight at 4 °C. GA^P^, GA^P^ + CO^P^, and GA^P^ × CO^P^ samples were added to the wells and incubated for 90 min. at 37 °C. The plate was washed four times with 1× PBS. The anti-human Fc fragment-specific IgG conjugated to HRP (Jackson) diluted at 1:1000 (100 μL) was used as the secondary antibody and added to the wells. The plates were incubated for 2 h at RT. After washing with 1 × PBS, each well was treated with 3,3′,5,5′-tetramethylbenzidine substrate solution (KPL) for 3 min. Absorbance was read at 450 nm while using a microplate reader (BioTek, Winooski, VT, USA). 

### 4.8. Transmission Electron Microscopy (TEM)

GA^P^ and CA^P^ × CO^P^ samples were resuspended in 20 μL PBS for TEM specimen preparation. Sample solution (5 μL) was loaded onto a carbon film-coated TEM grid that was rendered hydrophilic by glow discharge. After 90 s, excess sample solution was washed-off with distilled water. Uranyl acetate (1%; 5 μL) was loaded onto the grid for negative staining for 1 min., and the excess staining solution was blotted using a piece of filter paper. The samples were imaged using the JEM-1400Plus electron microscope (JEOL Ltd., Tokyo, Japan) equipped with a lanthanum hexaboride (Lab6) gun, operating at 120 kV. Images were recorded using an eight-megapixel, bottom-mount charge-coupled device (CCD) camera (EM-14650DR13; JEOL) [47].

### 4.9. Atomic Force Microscope (AFM)

AFM analysis was conducted using the NTEGRA Spectra system (NT-MDT; Santa Clara, CA, USA). Diluted solutions were allowed to down for 30 s spin coating at 400× *g*. GA^P^ and GA^P^ × CO^P^ protein sample solutions were applied to a freshly cleaved mica surface. To prepare GA^P^ and GA^P^ × CO^P^ samples on the functionalized substrate, the surface was ultrasonically washed with ethanol for approximately 1–3 min. to remove surface impurities. Treatment solution was applied to the mica surface and incubated for 15 min. at RT. The droplet was removed from the surface with nitrogen, and the surface was rinsed with water. The samples were analyzed using AFM. All of the measurements were conducted in air under ambient conditions.

### 4.10. Size Exclusion Chromatography-High Performance Liquid Chromatography (SEC-HPLC)

Analytical size exclusion chromatography-high performance liquid chromatogram phy (SEC-HPLC) was performed using an Agilent Bio SEC-5 (5 μm, 300 Å, 7.8 × 300 mm) column connected to an Agilent 1260 HPLC system (Agilent Technologies Inc., Santa Clara, CA, USA). The system and the column were equilibrated in 50 mM Na-phosphate pH 6.5, 300 mM NaCl at a flow rate of 1 mL/min. at 23 °C. The proteins were detected by monitoring absorbance at 280 nm. Advance Bio SEC 300A Protein Standard (Agilent Technologies Inc., Santa Clara, CA, USA) was used as a protein marker.

### 4.11. N-Glycan Analysis

Purified recombinant protein samples were first digested into glycopeptides using pepsin, as previously described [26,48]. From the glycopeptides, *N*-glycan was released using PNGase A (Roche, Basel, Switzerland). The released *N*-glycans were purified using graphitized carbon resin from Carbograph (Alltech). The purified glycans were re-dissolved in a mixture of 90 μL dimethyl sulfoxide (DMSO), 2.7 μL water, and 35 μL iodomethane for solid phase permethylation while using a spin column method [49]. The resulting permethylated glycans were mixed in equal volumes with 10 mg/mL 2,5-dihydroxybenzoic acid that was prepared in 1 mM of a sodium acetate solution. The mixture was applied onto a matrix-assisted laser-desorption-ionization (MALDI) MSP96 ground steel target plate and dried for MALDI-TOF mass spectrometry. All of the mass spectra were acquired at an acceleration voltage of 20 kV.

### 4.12. Immunological Analysis of Large Quaternary Recombinant Proteins (GA^P^ × CO^P^) in Mice

Six-week-old male BALB/c mice (six per group) were injected three times with 1 μg of GA^M^, GA^P^, GA^M^ + CO^M^, or GA^P^ × CO^P^ at two-week intervals. The negative control group was injected with 1 × PBS (100 μL). First, second, and third immunizations were conducted intraperitoneally (i.p.) without the adjuvant. Blood samples were collected by retro-orbital bleeding, 10 days after the third immunization. Mice were euthanized and bled by cardiac puncture. The sera were collected to confirm immune responses and to generate anti-GA IgGs from the immunized mice. GA^P^ proteins were immobilized on the GLC sensor chip (Bio-Rad). Mouse serum (10 μL) was applied onto the chip at pH 6 with a flow rate of 30 μL/min. at RT. SPR was performed, as described above.

### 4.13. DC and CD4^+^ T Cell Isolation

The spleens were aseptically excised from the immunized mice and placed in RPMI medium. Single cell suspensions were prepared. Collagenase D (400 U/mL; Roche, Basel, Switzerland) was added and the CD11c^+^ cells were enriched via MACS sorting (Miltenyi Biotech, Auburn, CA, USA). In brief, single cell suspensions from the spleens were prepared, and CD4^+^ T cells were enriched using MACS sorting [50]. All of the flow cytometry data were acquired using the BD FACS LSR II and analyzed by the FlowJo software (TreeStar, San Carlos, CA, USA).

### 4.14. Cytokine Assays

For the CD4^+^ T cell cytokine production assays, naive DCs were co-cultured with BALB/c T cells at a ratio of 1:10 in a 96-well U-bottom plate at 37 °C. After 72 h, the supernatants were collected, and the production of IL-4 and IL-10 was analyzed using the cytometric bead array (CBA) flex sets (BD Bioscience, San Jose, CA, USA) and flow cytometry [47,51].

### 4.15. Inhibition of Tumor Growth in Nude Mice

The SW620 human colorectal carcinoma cells (1 × 10^6^) were intradermally inoculated into the back of six-week-old BALB/c nu/nu mice (Japan SLC Inc., Hamamatsu, Shizuoka, Japan). After xenograft transplantation, six groups of mice were intraperitoneally administered with 40 μL of serum that was obtained from the 1× PBS, GA^M^, GA^P^, GA^M^ + CO^M^, or GA^P^ × CO^P^ immunized mice, four times at three-day intervals (totaling 160 μL over seven days). Positive control animals were injected with 100 μg CO^P^. Tumor growth was recorded at 8, 10, 12, and 15 days after initial injection while using graduated calipers and reported, as follows: Tumor volume (mm^3^) = width × length × height. The animal experiments were approved by the Institutional Animal Care and Use Committee (IACUC) at the Chung-Ang University, Seoul, Korea (approval ID: 14-0015, 26 August 2014). All of the methods were carried out in accordance with the approved guidelines.

### 4.16. Statistical Analysis

The results of the AAC challenge experiments were analyzed by ANOVA, followed by T-test for a single group-to-group comparison using Excel (Microsoft Corporation, Microsoft Office Excel 2013, Redmond, WA, USA). The differences were considered to be statistically significant when the *p*-value was <0.05.

## Figures and Tables

**Figure 1 ijms-21-05603-f001:**
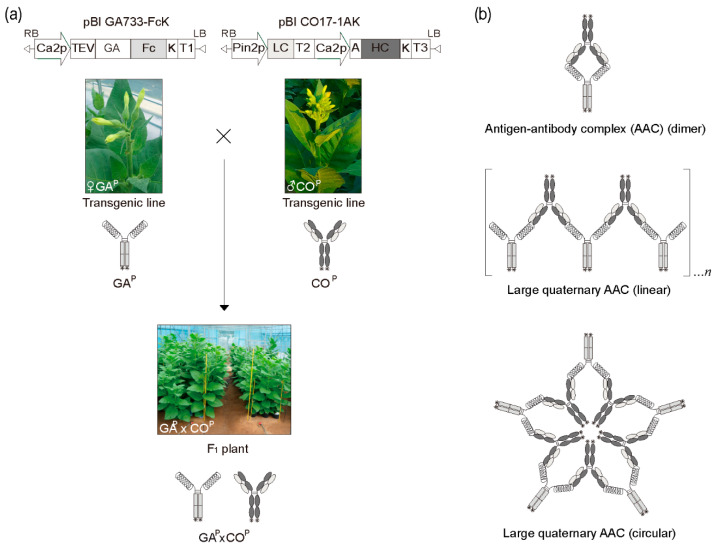
Schematic diagram of generation of F1 plants expressing both GA^P^ antigen and CO^P^ antibody by crossing transgenic plants expressing GA^P^ and CO^P^, and their different complex protein structures. (**a**) Plant vectors pBI GA733-FcK (left) and pBI CO17-1AK (right) for expression of GA^P^ and CO^P^ protein encoding genes in plants. F1 plants (GA^P^ × CO^P^) were generated by cross-fertilization of two different transgenic plants expressing GA^P^ and CO^P^. (**b**) Schematic diagram of different complex structures of both GA^P^ and CO^P^ in F1 plants. Dimers formed between solid GA^P^ and CO^P^ (Top). The linear chain structures between solid GA^P^ and CO^P^ (Middle). Large quaternary circular complex structures of GA^P^ and CO^P^ (Bottom). These structures are hypothetical.

**Figure 2 ijms-21-05603-f002:**
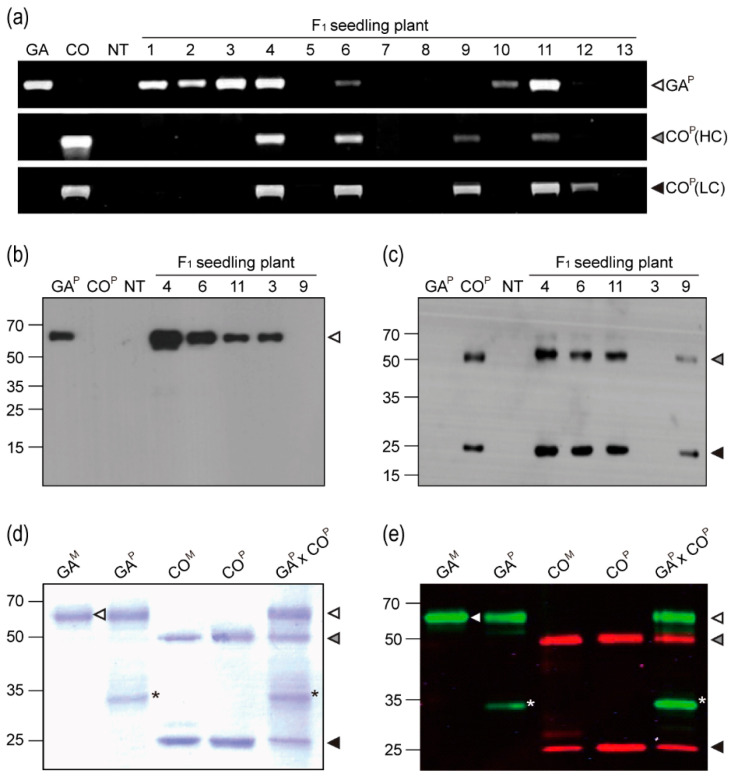
PCR and immunoblot analyses of GA^P^ × CO^P^ plants. (**a**) PCR was conducted to confirm the existence of the GA^P^ gene and the CO^P^ heavy chain (HC) and light chain (LC) genes in F_1_ plants. GA^P^, GA733-FcK; HC and LC, heavy and light chains of CO^P^, respectively; NT, non-transgenic plant. (**b**,**c**) Western blots were treated with mouse anti-GA IgG and anti-mouse Fc IgG conjugated to HRP to detect GA^P^; an anti-mouse IgG conjugated to HRP was used to detect CO^P^ HC and LC in total soluble proteins of F1 seedling plants, respectively. (**d**) SDS-PAGE analysis. Lanes: GA^M^, GA733-Fc from animal cell; GA^P^, GA733-Fc purified from GA plant; CO^M^, mAb CO17-1A, CO^P^, mAb CO17-1A purified from CO^P^ plants, GA^P^ × CO^P^, GA^P^ × CO^P^ purified from GA^P^ × CO^P^ plants. (**e**) Infrared fluorescence immunoblot to confirm the expression of both GA^P^ and CO^P^ in F_1_ plants. Black and grey arrows indicated GA^P^ and CO^P^, respectively. The asterisk indicates a non-specific protein band. The GA^P^ and CO^P^ bands were detected with a goat-human IRDye 800 cw (green) and an anti-mouse IRDye 680 LT (red), respectively. The white grey and black arrow heads indicate GA^P^, HC, and LC, respectively. Arrows on the blot indicate GA^M^.

**Figure 3 ijms-21-05603-f003:**
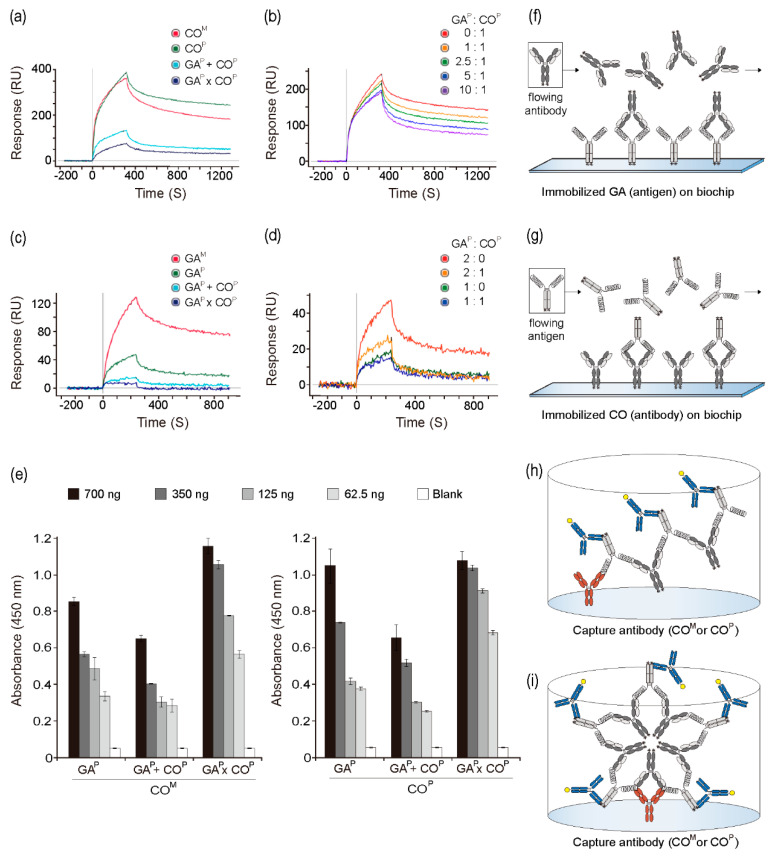
Interaction of GA^P^ × CO^P^ with anti-GA mAb (anti-colorectal cancer mAb CO17-1A) and plant-derived GA (GA^P^). Surface plasmon resonance was performed to confirm the interaction between GA^P^ and anti-GA mAb. (**a**) CO^M^, CO^P^, GA^P^ + CO^P^, and GA^P^ × CO^P^ samples were applied to the biochip fixed with the GA^P^. (**b**) The in vitro mixtures of GA^P^ and CO^P^ (with ratios of 0:1, 1:1, 2.5:1, 5:1, and 10:1) were applied to the biochip fixed with the GA^P^. (**c**) GA^M^, GA^P^, GA^P^ + CO^P^, and GA^P^ × CO^P^ were applied the biochip fixed with CO^P^. (**d**) The in vitro mixtures of GA^P^ and CO^P^ (in the ratios of 2:0, 2:1, 1:0, and 1:1) were applied to the biochip fixed with the CO^P^. (**e**) Confirmation of the binding activity of CO^M^ and CO^P^ to GA^P^ × CO^P^ by ELISA. The wells were coated with CO^M^ and CO^P^ as capture antibodies (50 ng/well). GA^P^, GA^P^ + CO^P^, and GA^P^ × CO^P^ were added to the wells. Anti-human Fc IgG conjugated to HRP was added as a detection antibody. (**f**) Schematic diagram of the interaction between flowing antibody and immobilized antigen on the biochip using SPR. (**g**) Schematic diagram of the interaction between flowing antigen and immobilized antibody on the biochip using SPR (**h**) Schematic diagram of the expected linear structure complex in plate. (**i**) Schematic diagram of the expected circular structure complex in plate.

**Figure 4 ijms-21-05603-f004:**
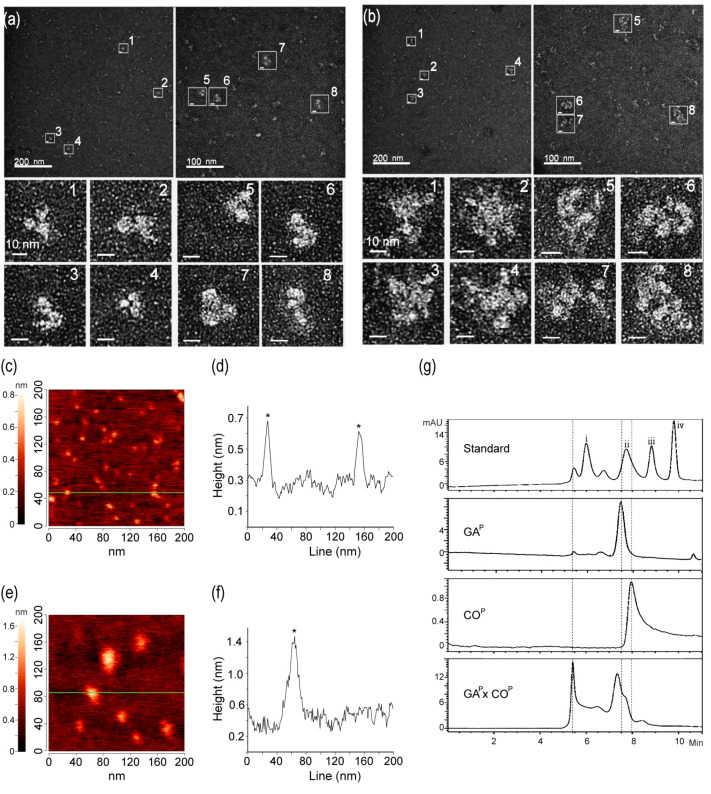
Characterization of GA^P^ and GA^P^ × CO^P^ protein structure and size. Protein structure and size analyses were performed by electron microscopy (EM) (**a**,**b**), atomic force microscopy (AFM) (**c**–**f**), and Size Exclusion Chromatography-High-Performance Liquid Chromatography (SEC-HPLC). (**a**) Y-shape structure of GA^P^. (**b**) Circular complex quaternary structures (multimerization) of GA^P^ × CO^P^. (**c**) Atomic force microscopy (AFM) of GA^P^ only, measuring 10–15 nm in diameter. (**d**) Asterisks on the line indicate GA^P^. (**e**) Atomic force microscopy (AFM) of GA^P^ × CO^P^, measuring 30~35 nm diameter; (**f**) an asterisk on the line indicates GA^P^ × CO^P^. (**g**) SEC-HPLC was conducted to determine the protein size of GA^P^, CO^P^, and GA^P^ × CO^P^. The dotted lines indicated main peaks of GA^P^, CO^P^, and GA^P^ × CO^P^. The i, ii, iii, iv mark in standard were 670, 150, 45, 17 kDa, respectively. IgM, IgM from human serum; GA^P^, plant-derived GA-Fc; CO^P^, plant-derived Anti-GA mAb; GA^P^ × CO^P^, plant-derived antigen-antibody complex (AAC).

**Figure 5 ijms-21-05603-f005:**
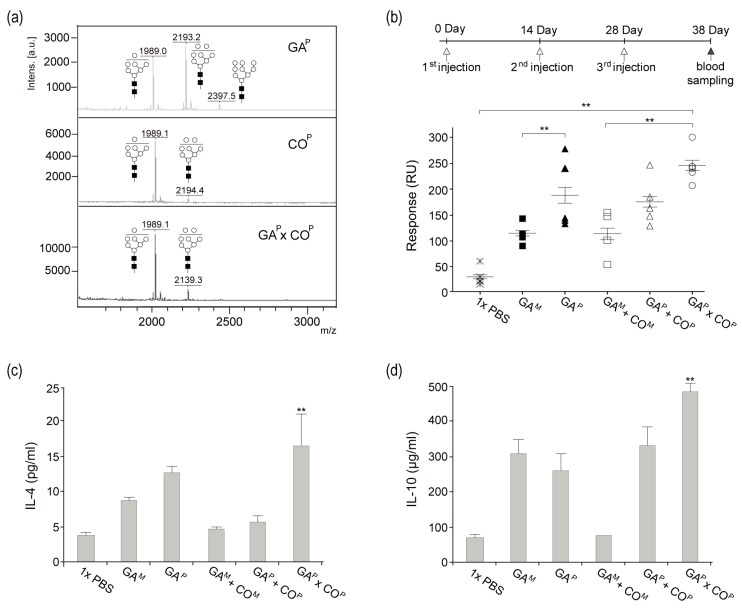
(**a**) *N*-glycosylation of GA^P^ × CO^P^ and induction of anti-GA IgG in mice injected with GA^M^, GA^P^, GA^M^ + CO^M^, GA^P^ + CO^P^, and GA^P^ × CO^P^. (**a**) *N*-glycosylation of GA^P^, CO^P^, and GA^P^ × CO^P^. The profiles of *N*-glycans from GA^P^, CO^P^, and GA^P^ × CO^P^ were analyzed using MALDI-TOF mass spectrometry. Black square, *N*-acetylglucosamine; white circle, mannose. (**b**) Six week old male BALB/c mice were maintained in a pathogen-free environment. Seven-week-old male BALB/c mice were injected three times with 3 μg of GA^M^, GA^P^, GA^M^ + CO^M^, GA^P^ + CO^P^, and GA^P^ × CO^P^, and blood samples were collected. Sera from immunized mice were determined to contain anti-GA mAbs through SPR with the GA^P^ fixed biochip. GA^M^, mammalian-derived GA733-Fc; GA^P^, GA733^P^-FcK; GA^M^ + CO^M^, in vitro mixture of GA^M^ and CO^M^; GA^P^ + CO^P^, in vitro mixture of GA^P^ and CO^P^; GA^P^ × CO^P^, GA^P^ × CO^P^ purified from plant. Individual counts and mean values (*n* = 5) for each group are shown. (**c**,**d**) Flow cytometric analysis of IL-4 and IL-10 in a CD4^+^ subset of Th2 cells. Naive DCs were co-cultured with immunized BALB/c T cells at a ratio of 1:10 (DC, 0.3 × 10^5^: T cell, 0.3 × 10^6^) in a 96-well U-bottom plate at 37 °C. After 72 h, supernatants were collected. IL-4 and IL-10 production was measured by a cytometric bead array (CBA) assay. The asterisks indicate statistically significant differences (** *p* < 0.01).

**Figure 6 ijms-21-05603-f006:**
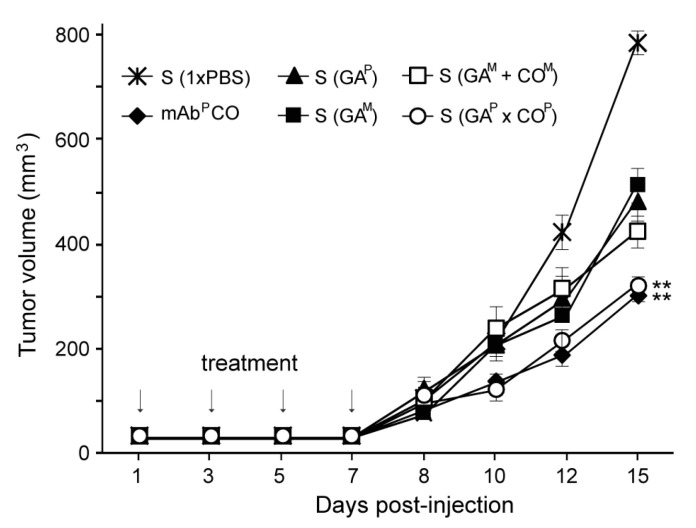
Suppression of tumor growth in nude mice through administration of sera from mice immunized with GA^P^ × CO^P^. BALB/c nu/nu mice were intradermally injected with 1 × 10^6^ SW620 cells. Mice were subsequently injected with serum obtained from mice immunized with 1× PBS, GA^M^, GA^P^, GA^M^ + CO^M^, or GA^P^ × CO^P^ (S (1× PBS), S (GA^M^), S (GA^P^), S (GA^M^ + CO^M^), or S (GA^P^ × CO^P^), respectively). The positive control was the CO^M^ antibody treated group. At day 1, 4, and 7, all mice were injected with three additional doses of serum or CO^M^ antibody. Tumor volumes (mm^3^) were recorded at 8, 10, 12, and 15 days after the initial inoculation of cancer cells. The asterisks indicate statistically significant differences (** *p* < 0.01).

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
