# Peer review of "A Plant-Derived Antigen–Antibody Complex Induces Anti-Cancer Immune Responses by Forming a Large Quaternary Structure"

_ijms, 2020, doi:10.3390/ijms21165603_

Round 1
Reviewer 1 Report
- Line 14; there is an 0 character in beneficial: be0neficial
- Line 20; remove double promising
- Line 28; in plant(s), antigen space fused or antigen-fusion
- Line 30; fix up
- You have 2 Figure 2s
- The EM photos are too low resolution to read. They need to be cleaned up quite a bit. I can not tell the overall structure at all, even when zoomed in – please clean up as this is a key figure in this paper
- Why are GAp COp complexes different than GAm and COm complexes? Why was it nessessary to use plants? How do these compare in EM? Do they not form complexes? Why not? I may have missed it, but I don’t see an answer to these questions? I see many comparisons of both groups, but none to answer why they are different – and why plant-derived complexes should be any different? If you take GA and CO expressed independently and mix them – do they not form complexes? Why not? All this is really fundamental to your premise.
- Last figure – tumor growth. When you discuss the results, be explicit in that you are attempting to induce a long-lasting immune response. In the discussion talk more about next steps to verify long-lasting immune response vs antibody treatment. I think this is the first step, but comes just-short of saying why this method will be better. We know it produces an immune response, but is this protective? How long does this last? I realize this is beyond the scope of the paper, but it would be good to discuss it. One would hope that it would help reduce metastatis, or require less injections to reduce the tumor, but discuss it. Could it be used as an anti-cancer vaccine as is? Why or why not? All should go in the discussion.
Author Response
Response to Reviewer 1 Comments
This is the revised manuscript entitled “A plant-derived antigen–antibody complex induces anticancer immune responses by forming a large quaternary structure” according to reviewers’ helpful comments and suggestion. We hope that the revised manuscript is qualified for publication in International Journal of Molecular Sciences.
Reviewer 1
- Line 14; there is an 0 character in beneficial: be0neficial
- We changed the word ‘be0neficial’ to ‘beneficial’. (Page 2 line 41)
- Line 20; remove double promising
- The sentence “as a promising alternative with promising product safety and economic benefit.” Has been changed to “as a promising alternative with product safety and economic benefit.” (Page 2 Line 47).
- Line 28; in plant(s), antigen space fused or antigen-fusion
- We changed the sentence ‘antigen-fused’ to ‘antigen fused’. (Page 2 line 56)
- Line 30; fix up
- The sentence “we would like to show another way to make immune complex form using transgenic plant crossing process.” is changed to “we demonstrated another way to make immune complex form using transgenic plant crossing process.” (Page 2 Line 57-58).
- You have 2 Figure 2s
- Thank you for the correction. We changed the figure number of manuscript correctly.
- The EM photos are too low resolution to read. They need to be cleaned up quite a bit. I cannot tell the overall structure at all, even when zoomed in – please clean up as this is a key figure in this paper
- I truly appreciate the reviewer comments. According to the reviewer’s comment, the EM photos of figure 4 with poor quality have been changed to the EM photos data with better resolution.
- Why are GAP´COP complexes different than GAM+COM complexes? Why was it necessary to use plants? How do these compare in EM? Do they not form complexes? Why not? I may have missed it, but I don’t see an answer to these questions? I see many comparisons of both groups, but none to answer why they are different – and why plant-derived complexes should be any different? If you take GA and CO expressed independently and mix them – do they not form complexes? Why not? All this is really fundamental to your premise.
- We authors assume that the structure of the in vitro mixture of GAM + COM is not different from that of the GAPxCOP F1 plant expressing both GAP and COP has better naïve conditions to have the AAC form of GAP x COP compared to the in vitro mixture of GAM + GOM since the GAP and COP are folded and assembled in plant ER where post-translation events occur such as protein folding, protein assembly, and glycosylation. Furthermore, co-expression of both GAP and COP in a single plant has another advantage that we don’t’ need an extra step for in vitro mixture process to generate ACC form. It only requires to grow plants and simply purify them. To determine whether the in vitro mixtures have a complex form, SPR with the mixtures was conducted as described in Figure 3. As shown in Figure 3, the mixtures had less binding activity compared to the in vitro mixture of GA and CO, which means the GA is bound to CO (anti-GA mAb) having ACC form. These data indicate that the naïve plant cell should render a better or stronger assembly of both GA and CO than the in vitro mixture process. In addition, ELISA was conducted to reconfirm the ACC assembly. We hope these answers can make clear about our current studies and logics based upon the current data. These points are in ‘Results’ and ‘Discussion’.
- Last figure – tumor growth. When you discuss the results, be explicit in that you are attempting to induce a long-lasting immune response. In the discussion talk more about next steps to verify long-lasting immune response vs antibody treatment. I think this is the first step, but comes just-short of saying why this method will be better. We know it produces an immune response, but is this protective? How long does this last? I realize this is beyond the scope of the paper, but it would be good to discuss it. One would hope that it would help reduce metastatic, or require less injections to reduce the tumor, but discuss it. Could it be used as an anti-cancer vaccine as is? Why or why not? All should go in the discussion.
- We authors fully agree with the reviewer’s point of view about the long last immune response. We hope that the GA x CO ACC complex vaccine candidate expressed in plant can be used for anticancer immunotherapy in cancer patient. In our current study, we showed the immune response to generate anticancer IgGs to inhibit colorectal cancer cell growth in animal model. However, we did not show the protective or preventive immune responses in animal model after vaccination of the GAPxCOP. In the future, it is required to determine protective immunity of GAPxCOP against cancer induced in animal model with its vaccination (Wei et al., 2015). In addition, in vitro migration or wounding healing assays would be essential to investigate the anti-metastatic activity of the anticancer IgGs induced in the vaccinated mice (Park et al., 2014; 2020).
- The discussion sentences described above are added in ‘Discussion’ part (Page 11 Line 339-345).
Park, S. R.; Ko, K.; Lim, S.; Cha, S. Y.; Chung, H. J.; Park, S. J.; Myung, S. C.; Kim, M. K., In vitro wound healing: Inhibition activity of insect‐derived mAb CO17‐1A in human colorectal cancer cell migration. Entomological Research 2020, 50, (4), 199-204.
Wei, W.-Z.; Jones, R. F.; Juhasz, C.; Gibson, H.; Veenstra, J., Evolution of animal models in cancer vaccine development. Vaccine 2015, 33, (51), 7401-7407.
Park, S. R.; Shin, Y. K.; Lee, K. J.; Lee, J. H.; Hedin, D.; Mulvania, T.; Lee, S. H.; Ko, K., Expression, glycosylation and function of recombinant anti‐colorectal cancer mAb CO17‐1A in SfSWT4 insect cells. Entomological Research 2014, 44, (1), 39-46.
Reviewer 2 Report
The manuscript has demonstrated that the serum from the AAC administrated mouse has a higher potential to inhibit the cancer growth in colorectal cancer xenograft nude. Here are some concerns list below:
Major:
- The manuscript has shown the higher antibody titer in the AAC administrated mouse, and the following serum transfer experiment has found a similar efficacy between mAB and the serum from AAC administrated mouse for inhibition of cancer growth. These data indicated the higher antibody production in AAC administrated mouse might play a major role to control the cancer growth. Thus, the main point of this manuscript is that the AAC is able to increase antibody production, but, not the anti-cancel immunity.
Minors:
- Line14, “be0neficial” should be beneficial.
- Are there any roles of higher cytokines (IL4, IL10) in the serum play for controlling cancer growth (need to discuss)?
- What is the antibody level (µg?) in the serum compared to the mAb (100µg)?
Author Response
Response to Reviewer 2 Comments
This is the revised manuscript entitled “A plant-derived antigen–antibody complex induces anticancer immune responses by forming a large quaternary structure” according to reviewers’ helpful comments and suggestion. We hope that the revised manuscript is qualified for publication in International Journal of Molecular Sciences.
Reviewer 2
The manuscript has demonstrated that the serum from the AAC administrated mouse has a higher potential to inhibit the cancer growth in colorectal cancer xenograft nude. Here are some concerns list below:
Major:
- The manuscript has shown the higher antibody titer in the AAC administrated mouse, and the following serum transfer experiment has found a similar efficacy between mAb and the serum from AAC administrated mouse for inhibition of cancer growth. These data indicated the higher antibody production in AAC administrated mouse might play a major role to control the cancer growth. Thus, the main point of this manuscript is that the AAC is able to increase antibody production, but, not the anti-cancer immunity.
- I truly appreciate important point of the current manuscript. We authors fully agree with the reviewer’s comments and suggestion. In Abstract, and Discussion, the main points that the AAC is able to increase antibody production are newly added as follows:
- Abstract: The words ‘increases anti-cancer antibody production’ are added after removal of the words ‘enhance immune response’. (Page 1 line 23-24). In addition, the entire ‘Abstract’ has been carefully reorganized and revised.
- Discussion: The words ‘an immune response’ are replaced with the words ‘production of anti-colorectal cancer antibody’. (Page 10 line 263)
Minors:
- Line14, “be0neficial” should be beneficial.
- We changed the word ‘be0neficial’ to ‘beneficial’. (Page 2 line 41)
- Are there any roles of higher cytokines (IL4, IL10) in the serum play for controlling cancer growth (need to discuss)?
- IL-4 and IL-10 are involved in inducing humoral immunity and play a role in inducing IgG1 antibodies, which mediate complement-dependent cytotoxicity (CDC) and recruit effector cells for antibody-dependent cellular cytotoxicity (ADCC) as a promising antibody isotype for tumor immunotherapy (Finkelman et al., 1990; Krestschmer et al., 2017).
- The above discussion sentences are newly included (Page 11 line 309-312).
Kretschmer, A.; Schwanbeck, R.; Valerius, T.; Rösner, T., Antibody isotypes for tumor immunotherapy. Transfusion Medicine and Hemotherapy 2017, 44, (5), 320-326.
Finkelman, F. D.; Holmes, J.; Katona, I. M.; Urban Jr, J. F.; Beckmann, M. P.; Park, L. S.; Schooley, K. A.; Coffman, R. L.; Mosmann, T. R.; Paul, W. E., Lymphokine control of in vivo immunoglobulin isotype selection. Annual review of immunology 1990, 8, (1), 303-333.
- What is the antibody level (µg?) in the serum compared to the mAb (100µg)?
- In general, the amount of anti-GA733 mAb has not been measured in serum (Lu et al., 2012; Fu et al., 2018). However, as shown in previous studies (Lu et al. 2012; Fu et al., 2018), the relative information was obtained to determine whether the antibody specifically recognizing antigens injected to mice. In our current study, we injected 160 μl of serum from the vaccinated mice to nude mice and 100 μg of anti-colorectal cancer mAb as a positive control. The 160 μl of serum showed similar inhibition of tumor growth to 100 μg of anti-GA733 mAbM. Thus, we assumed that there is at least decent amount of anti-GA733 IgGs having similar anti-tumor activity to 100 μg of anti-GA733 mAbM in 160 μl of serum.
Fu, Y.-Y.; Zhao, J.; Park, J.-H.; Choi, G.-W.; Park, K. Y.; Lee, Y. H.; Chung, I. S., Human colorectal cancer antigen GA733-2-Fc fused to endoplasmic reticulum retention motif KDEL enhances its immunotherapeutic effects. Journal of cancer research and therapeutics 2018, 14, (10), 748.
Lu, Z.; Lee, K.-J.; Shao, Y.; Lee, J.-H.; So, Y.; Choo, Y.-K.; Oh, D.-B.; Hwang, K.-A.; Oh, S. H.; Han, Y. S., Expression of GA733-Fc fusion protein as a vaccine candidate for colorectal cancer in transgenic plants. Journal of Biomedicine and Biotechnology 2012, 2012.